# Effects of a 6 Week Low-Dose Combined Resistance and Endurance Training on T Cells and Systemic Inflammation in the Elderly

**DOI:** 10.3390/cells10040843

**Published:** 2021-04-08

**Authors:** Michael Despeghel, Thomas Reichel, Johannes Zander, Karsten Krüger, Christopher Weyh

**Affiliations:** 1Despeghel and Partner Health Consulting, CH-8280 Kreuzlingen, Switzerland; despeghel@despeghel-partner.de; 2Department of Exercise Physiology and Sports Therapy, Institute of Sports Science, Justus-Liebig-University Gießen, 35394 Gießen, Germany; thomas.reichel@sport.uni-giessen.de (T.R.); christopher.weyh@sport.uni-giessen.de (C.W.); 3Laboratory Medical Care Centre Constance GmbH, 78467 Konstanz, Germany; j.zander@labor-brunner.de

**Keywords:** aging, exercise, immunosenescence, inflammation, inflammaging, resistance training, endurance training

## Abstract

With increasing age, the immune system undergoes a remodeling process, affecting the shift of T cell subpopulations and the development of chronic low-grade inflammation. Clinically, this is characterized by increased susceptibility to infections or development of several diseases. Since lifestyle factors can play a significant role in reducing the hallmarks of immune aging and inflammation, we investigated the effect of a 6 week low-dose combined resistance and endurance training program. Forty participants (70.3 ± 5.0 years) were randomly assigned to either a training (TG) or control group (CG) and performed a controlled low-threshold and care-oriented 6-week-long combined resistance and endurance training program. Changes in anthropometrics as well as strength capacity were measured. In subgroups of TG and CG, T cells and their subpopulations (CD4^+^, CD8^+^, naïve, central, effector memory, T-EMRA) were analyzed by flow cytometry. The changes of various plasma cytokines, chemokines, growth factors and adipokines were analyzed by luminex assays. The exercise program was followed by an increase in strength capacities. Participants of TG showed an increase of the CD4^+^/CD8^+^ T cell ratio over time (*p* < 0.05). Significant decreases in systemic levels of interleukin (IL-) 6, IL-8, IL-10 and vascular endothelial growth factor (VEGF) (*p* < 0.05) were observed for participants of TG over time. Even short-term and low-threshold training can reduce some of the hallmarks of immune aging in elderly and thus could be beneficial to stimulate immunity. The specific characteristics of the program make it easily accessible to older people, who may benefit in the longer term in terms of their immunocompetence.

## 1. Introduction

As people age, changes in all organs and tissues have been described, including the immune system. Several components of the immune system undergo a remodeling process, termed “immunosenescence”, which is associated with significant shifts in leukocyte populations and dysregulation of major immune functions [1]. Age-related changes affect both innate and adaptive immunity, with the T cell compartment showing the most marked senescence-associated processes. While the total number of T cells remains constant throughout life, the proportion of CD8^+^ cells increases. At the same time, there is a reduction in the number of CD4^+^ cells, leading to a decrease in the CD4^+^/CD8^+^ ratio. Within the CD4^+^ and CD8^+^ subpopulations, immune aging is associated with a decrease in cells with a naïve phenotype, while the proportion of highly differentiated memory cells increases [2,3]. Naïve T cells, which express CD45RA and CCR7, represent a pool of antigen-inexperienced cells that ensure an adequate immune response against newly invading pathogens. Progressive thymic involution and T cell differentiation, driven by repetitive antigen stimulation and inflammation, reduces the proportion of these T cell subtypes. Especially after the age of 65, a shift to a more senescent T cell subtype and an accumulation of highly differentiated T cells, which lack the expression of CCR7, has been described. Additionally, a specific highly differentiated T cell population, termed T-EMRA cells (terminal differentiated effector memory T cells re-expresses CD45RA) appear more frequently [3,4]. Further hallmarks of immunosenescence include the increase of late-differentiated T cells with lack of co-stimulatory molecule CD28 or expression of CD57 [5,6]. These cells are characterized by shortened telomeres, a reduced proliferative capacity and by the production of mainly pro-inflammatory cytokines [2,7]. Through the resulting association between immune-remodeling processes and the increase of systemic inflammatory cytokines, the term “inflammaging” was introduced [8]. Typically, a number of pro-inflammatory molecules like interleukin (IL) -1β, IL-6 or tumor necrosis factor-α (TNF-α) increase over lifespan and can, depending on their extent, force the aging process towards pathologic conditions [9,10].

The general importance of age-associated changes in the T cell repertoire is highlighted by their inclusion in the immune risk profile (IRP). The IRP represents a cluster of immunological parameters that have been shown to be associated with poor immune function in the elderly and are predictive of earlier mortality [3]. Clinically, it is characterized by increased susceptibility to infections, a more frequent reactivation of latent viruses and decreased vaccine efficacy [11,12,13]. Likewise, the immunosenescence and inflammaging could thus be part of the lethality amongst the elderly with COVID-19 [14]. Furthermore, increases in autoimmunity and cancer seems also be related to an aging immune system [15,16] and the facilitation of various internal, orthopedic, psychological and neurodegenerative conditions has been intensively discussed [17,18].

Many findings have proven that a physically active lifestyle can have positive effects on the aging immune system [19,20]. Results from cross-sectional studies report that trained older individuals show less hallmarks of immunosenescence compared to untrained peers [21]. Some findings from these data have been supported by controlled exercise interventions, while others have not been successfully replicated. Thus, after three weeks of endurance training in prediabetic participants, a proportional increase in CD4^+^/CD8^+^ ratio, naïve and central memory (CM) T cells was found, while the proportion of senescent CD8^+^ T-EMRA cells decreased at the same time [22]. Based on these initial findings, it would be important to investigate what kind of physical activity effectively influences the immune ageing process. Positive results have largely been observed for endurance training, while the effects of strength training have hardly been studied so far [23,24].

Therefore, the aim of the current study was to evaluate the effects of a 6-week-long combined low-dose resistance and endurance training on the T cell compartment and plasma level cytokines in a general population of older participants. For this study, it was taken into account that not every type of activity is accessible and feasible for older people. Accordingly, an exercise program was chosen which was close to care and low-threshold. Concerning the main findings of literature, we hypothesized that the training program would induce changes in T cell populations and in plasma cytokine levels compared to the control group.

## 2. Materials and Methods

### 2.1. Study Design

The present study was conducted as a pre–post randomized controlled trial. Considering a potential higher dropout rate during COVID-19 pandemic in the intervention group (risk of quarantine) and to gain a higher incentive for participation, participants were randomly allocated unequally in a ratio of 3:1 to one of the two study groups. We chose an unequal randomization of 3:1 because of the COVID-19 pandemic. On the one hand, it was supposed that an increased risk of quarantine could lead to a higher dropout rate in the intervention group. On the other hand, due to the was a risk of a second lockdown, recruitment should be accelerated. Evidence suggests the chance to be drawn into the intervention group increases the motivation to participate. In both cases, unequal randomization is a common tool [25]. Group 1 performed a 6-week combined strength and endurance training (training group = TG) and group 2 received no additional treatment (control group = CG). All measurements were conducted before training (T1) and after training (T2). 

### 2.2. Participants

Forty participants (70.3 ± 5.0 years) were randomly assigned to a training group (TG *n* = 30) or a control group (CG *n* = 10). Participants of the TG completed a 6-week-long intervention, while participants of the CG were encouraged to maintain their normal activity level. Their anthropometrics are shown in Table 1. Participants for this study were recruited via advertisements in local newspapers and public notice boards in the general population. Inclusion criteria were age ≥60 and ≤75 years, previously physically inactive, defined as no specific physical exercise performed outside of everyday activities for at least 2 years, stable weight for at least half a year (±5 kg) and being able to physically perform the exercise intervention. The aim was to evaluate the effects of this program under typical everyday conditions and with an age-typical population. For this reason, healthy participants and participants with pre-existing pathologies were included. The health status (pathologies and medication intake) of the participants are shown in Table 1. Exclusion criteria included excessive consumption of alcohol, severe cardiovascular disease or bronchial asthma, poorly controlled diabetes, acute inflammatory or febrile diseases and generally any clinical condition that is a contraindication to exercising. The local ethics committee approved this study. All participants gave written informed consent before enrolment.

### 2.3. Body Characteristics and Physical Capacity

Body fat (percentage (%)), visceral body fat (m^2^) and skeletal muscle mass (kg) were analyzed by bioelectric impedance analysis (BIA) (Biospace InBody 770, Seoul, South Korea). Maximum strength tests were performed to measure muscular capacity and to determine load for exercise training. Therefore, isokinetic tests were performed on the same equipment where the strength training took place. This was the Milon Premium Circuit. It consists of six different devices (leg extension, leg flexion, chest press, seated row, abdominal press, back extension) that included all major muscle groups. For determination of maximum strength, three trials were completed for each muscle group, with contractions lasting 5 s, separated by 60 s rest intervals. Participants were encouraged verbally to elicit their maximal effort and force was displayed on a visual display in real time providing immediate feedback. Peak maximum strengths (kg) were recorded and the highest of the three trials was used for further analysis.

### 2.4. Exercise Training

The training program was designed close to daily living. It can be classified as a low-dose intervention which followed exercise guidelines for the general population from the American College of Sports Medicine [26]. It was designed in an age-appropriate, machine-supported and time-effective manner in order to minimize the barriers to participation. The training program took place at a commercial fitness center. All participants received an introduction by a sports scientist. Thereafter they performed a self-motivated training program twice a week. Each session consisted of a five minute warm up followed by two passes of a strength circuit (Milon, Emersacker, Germany) and a 20 min endurance program. The strength training consisted of six machine-supported exercises as described above (Section 2.3). Every exercise was performed for 1 min each for 15 repetitions. In between each exercise, the participants had 30 s of rest. The strength equipment worked with a motor regulating the force, and force was generated by this electric motor and was adjustable in 1 kg steps. The electronic adjustment of the equipment created optimal force curves, as the force curve was uniform from the start to the end position. The muscle was trained concentrically and eccentrically, loaded twice at the same time. All device settings, such as seat and lever positions, as well as the setting of resistance, repetition and heart rate worked automatically with the chip card. The training was documented and analyzed later. During the initial training session, the 1RM test was performed as described above (Section 2.3). The best of the strength test was used to set the machines to 60% of the participants 1RM. For the subsequent 6 weeks, participants were encouraged to increase their training load independently. Once the training load was reached over the two sets with 15 repetitions, it was increased at the next training session. The endurance exercise consisted of a 20 min bout performed on bicycle ergometers and cross-trainers at a heart rate of 190 minus age. Endurance training intensity was independently monitored by participants wearing a heart rate monitor. The training intensity was according to recommendations for exercise among individuals with pre-existing pathologies [27]. Including the warm-up and rest periods, the training session could be completed in approximately 1 h. Compliance of the participants was assessed via a training log.

### 2.5. Isolation of Peripheral Blood Mononuclear Cells

Venous blood samples were taken from each participant between the hours of 08:00–10:00. Participants were asked not to exercise for 24 h prior to blood sampling, to eat as usual and to arrive fasting in the morning for blood sampling. Erythrocytes were lysed by lysis buffer (Ebioscience, San Dieg, CA, USA) and, after centrifugation, leukocytes were frozen in freezing medium (Bambanker, GC Lymphotec Inc., Tokyo, Japan) for cell phenotyping at −80 °C. Serum was collected and also frozen at −80 °C for cytokine analysis.

### 2.6. Flow Cytometry

Frozen PBMCs were thawed at 37 °C and washed twice in RPMI 1640 with heat-inactivated FBS 20% (Life Technologies, Paisley, Scotland, UK). Post washing, pelleted cells were resuspended in PBS (1 × 10^6^ cells/mL). Incubation expired in the dark at room temperature for 20 min with five microliters (μL) of different fluorescence-coupled antibodies, respectively (BioLegend Inc., San Diego, CA, USA). CD4^+^ and CD8^+^ T cells were determined to calculate the CD4^+^/CD8^+^ ratio. In addition, CD4^+^ and CD8^+^ T cells can be classified into four distinct subsets based on expression of cell surface markers, CCR7 and CD45RA: naive (CD45RA^+^ CCR7^+^), central memory (CD45RA^−^ CCR7^+^), effector memory (CD45RA^−^ CCR7^−^) and effector memory RA (CD45RA^+^ CCR7^−^). Therefore, resuspended cells were stained with following combinations of antibodies: anti-human CD4 FITC (clone SK3) and anti-human CD8 Alexa Fluor ^®^ 700 (clone SK1) to identify the CD4 and CD8 T cells, anti-human CCR7 Brilliant Violet 421 ^TM^ (clone G043H7) and anti-human CD45RA APC (clone HI-100) to identify subsets. Further phenotypes described as highly differentiated senescent T cells (CD28^−^CD57^+^) was detected using the following antibodies: anti-human CD28 PerCP/Cyanine5.5 (clone CD28.2), anti-human CD57 PE (clone HNK-1). The gating strategy used to identify T cell subsets was set as follows: at first, lymphocytes were gated on a forward scatter/side scatter (FSC/SSC) dot plot following gating on CD4 and CD8. In this gate, absolute count of CD4^+^ and CD8^+^ was performed to determine CD4^+^/CD8^+^ ratio. Combinations of surface markers were classified to define the T cell subpopulations as described above (Figure 1). All T cell subpopulations were determined as the percentage of CD4^+^ and CD8^+^ gate, respectively. Appropriate isotype controls were used for setting gates. For flow cytometry analysis, we used CytoFLEX S (Beckman Coulter, Brea, CA, USA) and Kaluza analysis software 2.1 (Beckman Coulter, Brea, CA, USA). Spectral overlap when using more than one color was corrected via compensation.

### 2.7. Multiplex ELISA Luminex Assay

Plasma levels of various cytokines, chemokines and adipokines were determined using a human Magnetic Luminex Assay (Bio-Techne, Abingdon, Oxon, UK) and a Magpix Luminex instrument (Luminex Corp, Austin, TX, USA) according to manufacturer’s instructions. The following parameters were determined with the corresponding assay ranges: CC-chemokine ligand (CCL) 2 (30.9–7500 pg/mL), CCL13 (4.94–1200 pg/mL), intercellular adhesion molecule 1 (ICAM-1) (7000–1,700,000 pg/mL), IL-1ra (28.8–7000 pg/mL), IL-2 (0.64–28.00 pg/mL), IL-6 (0.85–35.00 pg/mL), IL-8 (1.6–11.70 pg/mL), IL-10 (0.24–10.0 pg/mL), IL-18 (18.5–4500 pg/mL), leptin (494–120,000 pg/mL), resistin (53.5–13,000 pg/mL), TNF-α (8.23–2000 pg/mL) and vascular endothelial growth factor (VEGF) (8.23–2000 pg/mL). All measured values for the cytokines were within the assay range and the corresponding standard curve.

### 2.8. Statistical Analysis

Statistical analysis was performed by SPSS version 26 (IBM^®^ SPSS Statistics 24, IBM GmbH, Munich, Germany). First, a Kolmogorov–Smirnov test was used to test all data on normal distribution. A few parameters of plasma cytokine analysis were not normally distributed in the TG or in the GC group. For normally distributed data, parametric tests could be applied for further analysis. To determine the interaction of group × time point and time effect, a two-way ANOVA with repeated measures (2 × 2) was calculated. Paired t-tests were used for post hoc analysis to detect differences between time points within groups. With the exception of CXCL-13, IL-18 and resistin, all parameters of the plasma cytokine analysis were not normally distributed. Therefore, we opted for nonparametric testing for these variables (Wilcoxon test for effects over time; Mann–Whitney U-test for time x group effects, using the mean differences of T2 minus T1 values). Additionally, due to a possible unequal distribution of pathologies within both groups, we analyzed the differences between TG and CG at TP1. Due to the explorative nature of the investigation, no adjustment of the alpha error for multiple testing was considered between all outcome parameters. If not indicated otherwise, results are given as arithmetic mean ± standard deviation (SD). The level of statistical significance was set at *p* ≤ 0.05.

## 3. Results

All study participants successfully completed the predetermined number of 12 training sessions.

### 3.1. Effects of Training Intervention on Health Status and Physical Capacity

The two way repeated-measure ANOVA revealed a significant time (F_(1;37)_ = 8.317; *p* = 0.007) and time × group interaction (F_(1;37)_ = 8.317; *p* = 0.002) effect for back extension. Post-hoc analysis showed significant increase of strength form T1 and T2 in the TG (*p* < 0.001). Significant time effects were also revealed for all other muscle strength parameters (abdominal press (F_(1;37)_ = 9.343; *p* = 0.004), leg extension (F_(1;37)_ = 21.726; *p* < 0.001), leg flexion (F_(1;37)_ = 4.986; *p* = 0.032), chest press (F_(1;37)_ = 37.313; *p* < 0.001) and row (F_(1;37)_ = 55.214; *p* < 0.001)). Post-hoc analysis showed an increase for abdominal press from T1 to T2 in TG (*p* < 0.001). Leg extension increased from T1 to T2 within TG (*p* < 0.001) as well as CG (*p* = 0.007). For leg flexion we revealed only an increase within TG (*p* = 0.006). Chest press revealed increase from T1 to T2 in TG (<0.001) and in CG (*p* = 0.003) as well as row increase from T1 to T2 in TG (<0.001) and in CG (*p* = 0.01) (Table 2). Additionally, we found a significant time effect for diastolic BP (F_(1;37)_ = 16.183; *p* < 0.001). Analysis points toward a decrease from T1 to T2 in TG (*p* < 0.001) as well as CG (*p* = 0.036) (Table 2).

### 3.2. Effects of Training Intervention on T Cell Subpopulations

The results of T cells and their subpopulations were measured among a subgroup of *n* = 16 participants. The two-way repeated-measure ANOVA revealed a significant time effect for ratio of CD4^+^/CD8^+^ (F_(1;13)_ = 6.923; *p* = 0.021). Analysis within groups showed significant increase of the ratio between T1 and T2 (*p* = 0.043) in the TG. Results in CG revealed no change (*p* = 0.241). Two-way ANOVA showed a significant decrease of the relative proportion of CD8^+^ EM T cells in CG from T1 to T2 (*p* < 0.001) over time (F_(1;13)_ = 5.113; *p* = 0.042). We also found a time effect for CD8^+^CD28^−^CD57^+^ T cells (F_(1;13)_ = 5.029; *p* = 0.046) and an increase of proportions within CG from T1 to T2 (*p* = 0.006) (Table 3). No changes were observed for the other T cell subpopulations.

### 3.3. Effects of Training Intervention on Plasma Cytokine Levels

IL-2 (z = −3528; *p* < 0.001; *n* = 31) was found to be decreased from T1 to T2 in TG (*p* = 0.003) as well as in CG (*p* = 0.027). Furthermore, we also found a time effect for IL-6 (z = −2665; *p* = 0.008; *n* = 40), IL-8 (z = −2965; *p* = 0.003; *n* = 40), IL-10 (z = −2259; *p* = 0.024; *n* = 33) and VEGF (z = −2312; *p* = 0.021; *n* = 40). Within all plasma cytokine levels, we revealed a decrease from T1 to T2 in TG (IL-6: *p* = 0.038, IL-8: *p* = 0.002, IL-10: *p* = 0.012, VEGF: *p* = 0.011) (Table 4).

## 4. Discussion

The present study evaluated the effects of a 6-week-long combined strength and endurance training program at a low volume and intensity in the elderly. The participants were a typical age-matched cohort, with age-appropriate comorbidities and medication. The results demonstrate that the exercise program induced an increase of the CD4+/CD8+ ratio in the present cohort. Statistically significant findings were also observed for the T cell subpopulations CD8^+^ EM and CD8^+^ CD28^−^CD57^+^. Analyses of plasma cytokines, chemokines, adiponectins and growth factors showed decreases in systemic levels of IL-6, IL-8, IL-10 and VEGF within the training group, indicating an immune-regulating effect of the exercise program. An improvement in strength capacity was observed in the majority of the training group, but differences in the control group were noted only for back extensor. Similarly, no interaction of group and time could be determined for any other parameter.

A number of previous longitudinal studies have identified the CD4^+^/CD8^+^ ratio as part of the IRP [3]. However, the normal CD4^+^/CD8^+^ ratio is poorly defined in healthy individuals. Ratios between 1.5 and 2.5 are generally considered; however, there is wide heterogeneity [28]. A decrease in the CD4^+^/CD8^+^ ratio is typically described with increasing age and is therefore considered as an accepted hallmark of immunosenescence. This is also applicable to chronic diseases associated with systemic low-grade inflammation [29]. The present study demonstrates that the regular strength and endurance training was able to increase in the CD4^+^/CD8^+^ ratio in elderly. Since the baseline CD4^+^/CD8^+^ ratio seemed to be slightly under range, the increase could be considered as a normalization. What is new about this study, however, is that this is already possible with a very low-threshold program within 6 weeks. In most short term exercise interventions in elderly, no changes in the CD4^+^/CD8^+^ ratio in were found [30,31,32]. Therefore, the sports program implemented here seems to have been immunologically very effective in this respect. Nevertheless, while the study design does not allow drawing any clinical conclusions from the changes it seems to restore an altered immune homeostasis in some way. However, as we have a slight but not significant increase in absolute CD4^+^ and CD8^+^ cells, we cannot exclude the possibility that other cell types also influence the changes in CD4^+^/CD8^+^ ratio. Since we do not used CD3 antibody to identify T cells, the change in CD4^+^/CD8^+^ ratio for the TG could because of absolute numerical changes in other cells that express CD4^+^ and CD8^+^ within the lymphocyte pool.

We further observed changes within CD8^+^ T cell subpopulations, indicated by altered proportions of CD8+ EM T cells and CD8^+^ CD28^−^CD57^+^ T cells. Aging is known to be accompanied by a decrease in frequency of naïve and an increase in the frequency of memory T cells, T-EMRA cells as well as an accumulation of senescent CD28^−^CD57^+^ T cells, with the effects most marked in the CD8^+^ pool [33]. Philippe et al. found proportional increases of CD4^+^ central memory cells, CD8+ naïve and central memory cells while proportions of CD4^+^ and CD8^+^ T-EMRA cells decreased after training [22]. In our study, the training program was not able to affect changes in frequency of CD4^+^ or CD8^+^ T cell subpopulations. It is assumed that the training was too short or too low-threshold for this, so that the stimulus was not strong enough. Differences are mainly observed in cross-sectional studies comparing a trained with an untrained collective in old age. Here, a long training history often provides for corresponding differences [21,34]. Our results only revealed increase of CD8^+^ EM T cells and decrease of CD8^+^ CD28^−^CD57^+^ T within CG. We are not quite sure why this happened in such a short time. In the exercise group, this change did not take place, which in turn makes a stabilizing effect on immune ageing possible. However, we do not want to over interpret the findings, as the control group was significantly smaller than the intervention group.

Immunosenescence is accompanied by a systemic increase in various pro-inflammatory molecules. The best-characterized cytokines are IL-6 and TNF-α. Regular exercise can affect this processes, indicated by a decrease of basal levels of IL-6 as well as TNF-α [35,36,37]. In the present study, however, this could only be observed within the training group for IL-6. This is not surprising, because other studies have also shown that TNF-α can mainly be reduced by exercise when there is already a significant increase in the baseline, e.g., in collectives with previous inflammatory diseases. For this, it is reported that the influences of exercise on IL-6 and TNF-α levels appeared to be more profound in healthy older adults than in those with a given disease or pathological condition [35]. Due to 67.5%of our participants are under pathological conditions, the effects of exercise might be less powerful. We also observed a decrease in the pro-inflammatory cytokine IL-8 within the exercise group. There are contradictory results in the literature about the effect of exercise programs on systemic IL-8. [38,39]. In this context, it can be speculated that greater effects can be expected with increasing intensity of the intervention [40]. IL-8 is a chemokine, which on the one hand attracts mainly neutrophilic granulocytes at the site of inflammation and on the other hand has a pro-angiogenic effect. In line with IL-6, lowering of IL-8 could be interpreted as a decrease in the status of systemic low-grade inflammation. Otherwise, IL-8 is produced and secreted in endothelial cells by mechanical stretching. Therefore, improved vascular function due to exercise might partly explain the inhibitory effect on this chemokine [41]. Furthermore, it could be shown that there is a relationship between IL-8 and IL-2 in the overweight [42], which could be in line with our results. According to this, we observed a decrease in plasma IL-2 levels within both groups. IL-2 could be characterized as an inflammatory cytokine, as it is associated with markers of inflammation as well as insulin resistance in overweight and obese individuals [42]. Therefore, decreases after exercise training could be related to lowering in chronic low-grade inflammation and are consistent with our findings for Il-6 and IL-8. We are not sure why the IL-2 levels in the CG were also somewhat reduced. It is possible that the participation of the test persons in the study led to somewhat more activity in the CG as well. This would also explain why the participants in the control groups also showed strength gains in some areas.

IL-10 is an immune regulatory cytokine with anti-inflammatory potential. Surprisingly, levels of plasma IL-10 were reduced in TG, while most studies showed no changes or an increase in IL-10 after regular exercise [43]. However, individuals with pathological conditions, such as obesity, reveal elevated IL-10 basal levels [44]. Therefore, in line with the decrease of pro-inflammatory IL-2, IL-6 and IL-8, a decrease in IL-10 could be related to a smaller release, since a lower counter-regulation of anti-inflammatory cytokines is necessary.

Supporting these findings, we observed a decrease in VEGF plasma levels within TG. Chronically elevated VEGF may be considered a pathological process as can be seen in tumor proliferative processes [45] or patients with chronic low-grade inflammation-associated conditions such as diabetes [46] or rheumatic diseases [47]. While acute exercise has been repeatedly shown to increase VEGF levels as a mediator of vascular adaptation, most studies on regular exercise programs did not find any changes of VEGF levels in healthy individuals [45]. The decrease in our study might reflect a downregulation in the sort of individuals having metabolic or cardiovascular risk factors.

In addition to the immunological effects, the changes in other parameters should not be ignored. No changes were found for many other proinflammatory factors. We assume that the intervention period was too short for this. In other studies, for example, changes in IL-18 were often only seen after 12 weeks [48]. A particular advantage of the training program used here was its low volume and low intensity, which will certainly increase access to exercise training for older individuals, while at the same time making it easier for them to do it on a regular basis.

The training succeeded in increasing strength. All strength tests showed an increase within the TG. Surprisingly, there were some improvements within the CG too, which could be explained by a learning effect due to isokinetic testing [49]. The leisure-time physical activity could also have an influence on the results of CG. Since we did not restrict them, it could be monitored in future studies. No increase of muscle mass was found, indicating that the improvement in strength capacity can be attributed to neuronal adaptations. Neither fat mass nor visceral fat could be reduced. The lack of distinct training effects on physical capacity or body composition could be in line with the underling results on hallmarks of immunosenescence or inflammatory markers. Therefore, it cannot be excluded that more significant improvements in strength or body composition could possibly promote more powerful impacts on immune system. Since the effects of endurance training are described in the majority to affect the immune system or inflammatory levels [20], variables of this training modality should be discussed in more depth. A progression was not performed within six weeks of training. Overall, the training period could be too short. As expected, longer training periods show greater adaptations [26].

Our study has several limitations. An examination of the endurance capacity did not taken place. Therefore, we cannot make any statements about the effectiveness of the training program on this point. Due to local conditions, we were unable to measure complete leukocyte counts and collect T cell subpopulation data for all subjects. In addition, we did not asses cytomegalovirus (CMV) serostatus in our study. CMV is associated with a greater number and proportion of senescent T cells [50]. In addition, CMV serostatus has been found to influence the magnitude and kinetics of CD8^+^ memory T cell responses to exercise [51,52]. Unequal randomization has a limiting effect on the power of statistical analyses. In addition, due to the explorative nature of the study, no sample size calculation was performed. The performance of a subgroup analysis among a reduced sample size must be regard as a limitation too. Therefore, results should be interpreted with care. Underlying diseases or disorders and medication intake that affect the immune system may lead to incorrect findings. However, we made a conscious decision to evaluate this training program among a typically aged population, such as would be found in a gym.

In conclusion, the current study demonstrates that a 6 week low-dose combined resistance and endurance training program seems to be effective in affecting hallmarks of immunosenescence and inflammaging among elderly individuals. These changes could be beneficial for a large group of the elderly, especially those with an immunological risk profile. The results might represent an opposite trend to aging of immune system and could thus stimulate immunity to pathogens and reduce severe viral disease. Due to the close to care and low-threshold characteristics of the training program, a large number of people could use it in a regular way. A long-term training approach with the implementation and evaluation of a systematic endurance program could achieve larger effects and should therefore be the subject of future research.

## Figures and Tables

**Figure 1 cells-10-00843-f001:**
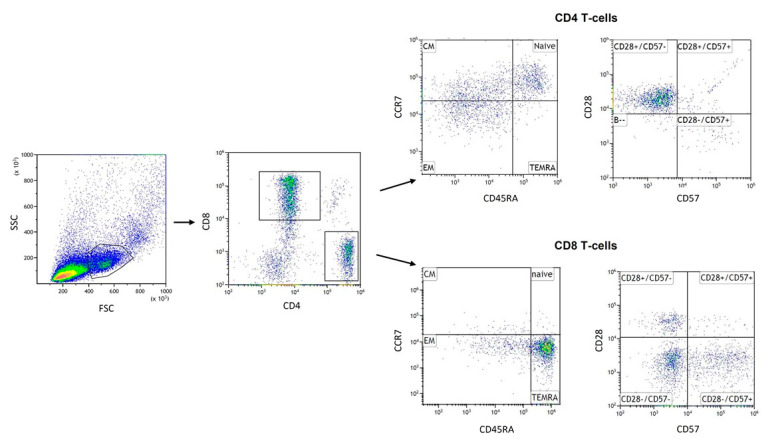
Gating strategy: Lymphocytes were gated in a FSC/SSC (forward scatter/side scatter) dot plot. From the cell population, CD4^+^ T cells and CD8^+^ T cells were gated according to their specific surface markers. T cell subpopulations, within CD4 and CD8 T cells, respectively, were defined by staining CD45RA and CCR7. Naïve (CD45RA^+^/CCR7^+^): naive T cells; CM (CD45RA^−^/CCR7^+^): central memory; EM (CD45RA^−^/CCR7^−^): effector memory; TEMRA (CD45RA^+^/CCR7^−^): effector memory + RA and late-differentiated T cells (CD28^−^/CD57^+^).

**Table 1 cells-10-00843-t001:** Anthropometric and health status of Training Group (TG) and Control Group (CG) (mean ± SD).

	TG	CG
*n* (male)	30 (14)	10 (7)
Anthropometrics		
Age (years)	70.4 ± 5.3	69.8 ± 4.4
Height (cm)	170.0 ± 7.3	171.9 ± 7.3
Weight (kg)	80.1 ± 11.7	77.1 ± 15.8
BMI (kg/m^2^)	27.7 ± 3.4	25.9 ± 3.5
Body fat (%)	31.6 ± 7.4	25.4 ± 8.6
Health status n (%)		
Participants with no pathologies	10 (33.3)	3 (30)
Participants with one pathology	10 (33.3)	4 (40)
Participats with pathologies ≥2	10 (33.3)	3 (30)
Pathologies n (%)		
Hypertension	9 (30)	5 (50)
Cardiovascular disease	3 (10)	1 (10)
Asthma	2 (13.7)	0 (0)
Hypercholesterolemia	3 (10)	1 (10)
Thyroid Disease	4 (13.3)	1 (10)
Mental disease	2 (6.7)	0 (0)
Other ^a^	7 (23.3)	2 (20)
Status of medicine intake n (%)		
Participants with no medicine	10 (33.3)	3 (30)
Participants with one medicine	9 (30)	4 (40)
Participats with medicine ≥ 2	11 (36.7)	3 (30)
Use of prescribed medicine n (%)		
Anti-hypertensive drugs	11 (36.7)	5 (50)
Anticoagulants	5 (16.7)	2 (20)
Statins	2 (6.7)	1 (10)
Inhaled sympathomimetics	2 (6.7)	0 (0)
Thyroid drugs	4 (13.3)	1 (10)
Psychotropic drugs	3 (10)	0 (0)
Other ^a^	5 (16.7)	2 (20)

^a^ Participants who were diagnosed and treated for rheumatoid arthritis, prostate disease, gout, sleep apnea.

**Table 2 cells-10-00843-t002:** Anthropometrics, health status and physical capacity parameters at time point 1 (T1) and T2 in training group (TG) and control group (CG) (mean ± SD).

	TG		CG		Time Effect	Time × Group Interaction
	T1	T2	T1	T2	*p*	*p*
**BMI (kg/m^2^)**	27.7 ± 3.5	27.7 ± 3.6	25.9 ± 3.5	25.8 ± 3.4	0.627	0.769
**Visceral fat mass (m^2^)**	123.4 ± 41.6	124.6 ±44.2	97.0 ± 36.7	101.3 ± 39.9	0.142	0.390
**Body fat (%)**	31.6 ± 7.4	31.6 ± 7.5	25.4 ± 8.6	26.3 ± 9.0	0.299	0.307
**Sceletal muscle mas (kg)**	30.1 ± 5.4	30.1 ± 5.3	31.7 ± 73	31.1 ± 6.8	0.112	0.139
**Back extension (kg)**	46.77 ± 17.4	65.00 ± 20.8 *	50.33 ± 22.4	49.11 ± 19.3	**0.007**	**0.002**
**Abdominal crunch (kg)**	30.73 ± 11.8	37.3 ± 11.9 *	34.33 ± 8.1	37.00 ± 13.4	**0.004**	0.205
**Leg extension (kg)**	56.4 ± 21.9	65.5 ± 21.6 *	51.7 ± 15.8	63.9 ± 18.1 *	**≤0.001**	0.218
**Leg flexion (kg)**	33.7 ± 16.0	38.5 ± 16.0 *	32.4 ± 15.5	35.2 ± 14.5	**0.032**	0.499
**Chest press (kg)**	50.9 ± 22.0	64.4 ± 23.1 *	54.3 ± 17.2	65.3 ± 18.9 *	**≤0.001**	0.555
**Seated row (kg)**	55.1 ± 19.2	66.5 ± 18.3 *	62.8 ± 19.8	70.9 ± 24.7 *	**≤0.001**	0.542

***** differences from T1 to T2 *p* ≤ 0.05.

**Table 3 cells-10-00843-t003:** CD4^+^ and CD8^+^ cells and their subpopulations at time point 1 (T1) and T2 in training group (TG) and control group (CG) (mean ± SD) among a subgroup of *n* = 16 participants.

	TG		CG		Time Effect	Time × Group Interaction
	T1	T2	T1	T2	*p*	*p*
**T cells** (cells/μL)						
**CD4** ^+^	431.63 ± 629.0	819.88 ± 1049.5	437.10 ± 275.3	964.43 ± 962.4	0.088	0.989
CD8^+^	443.88 ± 705.6	541.38 ± 855.2	410.29 ± 376.4	685.86 ± 528.4	0.394	0.463
CD4^+^/CD8^+^ ratio	1.15 ± 0.4	2.27 ± 1.5 *	1.20 ± 0.5	1.44 ± 0.7	0.021	0.109
T cell subsets (%)						
CD4^+^ naive	27.94 ± 6.5	26.52 ± 12.6	24.50 ± 9.2	25.31 ± 13.2	0.905	0.659
CD4^+^ CM	34.02 ± 7.0	38.15 ± 7.8	43.50 ± 10.7	39.92 ± 9.4	0.894	0.094
CD4^+^ EM	35.21 ± 11.2	33.74 ± 14.9	28.27 ± 6.7	30.97 ± 8.3	0.859	0.549
CD4^+^ T-EMRA	1.25 ± 0.7	1.52 ± 0.8	3.85 ± 3.1	3.84 ± 4.4	0.783	0.765
CD4^+^ CD28^−^CD57^+^	2.04 ± 2.1	2.37 ± 2.9	6.00 ± 6.0	5.69 ± 5.7	0.982	0.636
CD8^+^ naive	6.25 ± 3.1	5.48 ± 3.6	5.82 ± 3.1	6.93 ± 4.0	0.362	0.201
CD8^+^ CM	9.30 ± 5.4	6.72 ± 5.3	8.64 ± 5.1	9.29 ± 4.1	0.645	0.441
CD8^+^ EM	37.59 ± 16.4	35.20 ± 10.3	39.89 ± 12.7	29.56 ± 11.3 *	**0.042**	0.183
CD8^+^ TEMRA	46.90 ± 10.7	50.35 ± 15.3	48.35 ± 11.3	54.03 ± 9.3	0.143	0.715
CD8^+^ CD28^−^CD57^+^	40.48 ± 13.2	41.44 ± 11.0	35.00 ± 13.7	44.16 ± 16.7 *	**0.046**	0.096

***** differences from T1 to T2 *p* ≤ 0.05.

**Table 4 cells-10-00843-t004:** Cytokine plasma levels at time point 1 (T1) and T2 in training group (TG) and control group (CG) (mean ± SD).

	TG		CG		Time Effect	Time × Group Interaction
Molecules (pg/mL)	T1	T2	T1	T2	*p*	*p*
**CCL-2**	210.33 ± 91.4	190.20 ± 55.7	216.10 ± 83.6	212.37 ± 128.9	0.276	0.579
**CXCL-13**	33.17 ± 11.9	28.89 ± 11.7	44.51 ± 29.5	44.89 ± 31.4	0.198	0.126
**ICAM-1**	284,625.45 ± 135410.5	281,453.53 ± 112,499.9	354,343.68 ± 204,000.2	371,220.30 ± 168,159.4	0.610	0.601
**IL-1α**	3.96 ± 1.2	3.77 ± 1.1	3.69 ± 1.4	3.32 ± 1.0	0.376	0.646
**IL-1ra**	315.43 ± 189.2	280.59 ± 152.8	298.25 ± 124.3	266.30 ± 106.0	0.056	0.962
**IL-2**	3.69 ± 2.0	1.98 ± 1.7 *	2.80 ± 1.3	0.83 ± 0.7 *	**0.000**	0.789
**IL-6**	1.13 ± 0.3	0.98 ± 0.2 *	1.17 ± 0.5	1.03 ± 0.4	**0.008**	0.862
**IL-8**	3.05 ± 1.5	2.34 ± 0.7 *	2.60 ± 0.8	2.45 ± 1.1	**0.003**	0.456
**IL-10**	1.01 ± 0.4	0.87 ± 0.3 *	1.07 ± 0.4	1.04 ± 0.2	**0.024**	0.486
**IL-18**	179.71 ± 86.4	180.96 ± 80.5	216.97 ± 73.8	207.25 ± 60.1	0.640	0.544
**Leptin**	12,180.74 ± 11,654.7	14,804.82 ± 17,771.0	6095.97 ± 6350.4	6862.91 ± 10161.0	0.248	0.365
**Resistin**	7593.69 ± 2885.4	7634.37 ± 3263.1	7445.41 ± 1668.5	7770.8 ± 2068.7	0.477	0.580
**TNF-α**	2.55 ± 0.6	2.47 ± 0.5	2.49 ± 0.4	2.44 ± 0.5	0.423	0.646
**VEGF**	22.06 ± 16.1	14.99 ± 6.5 *	15.49 ± 5.9	15.49 ± 7.0	**0.021**	0.232

***** differences from T1 to T2 *p* ≤ 0.05.

## Data Availability

The data presented in this study are available in the article.

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
