# Peer review of "Effects of a 6 Week Low-Dose Combined Resistance and Endurance Training on T Cells and Systemic Inflammation in the Elderly"

_cells, 2021, doi:10.3390/cells10040843_

Round 1
Reviewer 1 Report
The manuscript “Effects of a 6-week low-dose combined resistance and endurance training on T cells and systemic inflammation in elderly” aimed to assess the effects of a low-dose exercise program on hallmarks of immune aging and inflammation. The study is well designed, and has an aspect of novelty in that it used a low-dose exercise program in older adults with many comorbidities rather than overly healthy older adults. The authors suggest that the exercise program improved physical strength, signs of immune aging and reduced inflammation. I have listed below some key comments that require attention before the manuscript is ready for publication.
Main Points
The first point below is the most important one for the authors to answer. Once this is made clear and discussed correctly, the additional points can be considered.
- The first F value presented in line 221 [F(1,13)] suggests that the immune analyses were only completed on 16 participants. I might be missing this, but I am unable to find an explanation of whether this was the case, or whether the immune analyses were completed on all participants, or if this F value is incorrect? If the immune analyses were only completed on a subset of participants, then this needs to be clearly defined and discussed. Further, it needs to be made clear, probably with another table, what the differences in physiology and inflammation between these 16 and the other participants were. You may want to consider revising the manuscript to show only the exercise results for these 16 participants as the CD4/CD8 ratio was your primary outcome.
- Please clarify how many of the total participants in each group had either a pathology or not. It looks like 30 people in the TG and 10 in the CG had a pathology, but it is unclear if this means everyone had a pathology or did some people have more than one pathology. In addition, the same is required for medications. Depending on how many participants in the whole study had a pathology you may need to describe differences between those with a pathology and those without at baseline.
- How were the groups stratified for randomization and who performed the randomization?
- More details are required for the blood draws please.
- Were participants fasted before each blood draw, and were lifestyles such as diet and exercise controlled in the 24-hour period before blood draws?
- It is unclear how the absolute T-cell counts were calculated, and to be clear was it the absolute counts or the percentages of CD4 and CD8 T-cells that were used to calculate the CD4/CD8 ratio?
- If the absolute counts were calculated as a percentage from a complete blood count this needs to be described. Also, if a CBC was completed it would be useful to include the counts of total WBCs, neutrophils, monocytes and lymphocytes.
- At T2 there is a large mean increase, albeit non-significant (p=0.088), in absolute CD4+ T-cell counts and a small mean increase for absolute CD8+ T-cell counts for both the TG and CG. This is the likely cause of the change in the ratio. However, the changes in frequencies of T-cell subsets do not reflect these differences. Specifically, although the CD4/CD8 ratio increases there is no beneficial change in the CD4 or CD8 subset frequencies. Therefore, could the observed change in CD4/CD8 for the TG be because of absolute numerical changes in other cells that express CD4 (e.g. dendritic cells) and CD8 (e.g. NK-cells, MAITs, DCs) within the lymphocyte pool? Critically, because you have not used CD3 to identify T-cells, you may have inadvertently included other cells – this should be discussed.
- For the cytokine analyses, please report the lower limit of detection (LLOD) and what percentage of each cytokine was within the LLOD
- Given my above comments, the limitations and conclusion section may need revised.
Grammar – there are a few errors throughout that I ‘m sure the journal will pick up, but to help I have highlighted a few.
Line 223. p=0,241 needs changed to p=0.241
Line 319. State how many “a large proportion” is.
Line 322. Change sports programmes to exercise programmes
Line 371. Change discuss to discussed
Table 2. Change mas to mass
Throughout, aging and ageing have both been used – please be consistent.
Author Response
Reviewer 1
- The first F value presented in line 221 [F(1,13)] suggests that the immune analyses were only completed on 16 participants. I might be missing this, but I am unable to find an explanation of whether this was the case, or whether the immune analyses were completed on all participants, or if this F value is incorrect? If the immune analyses were only completed on a subset of participants, then this needs to be clearly defined and discussed. Further, it needs to be made clear, probably with another table, what the differences in physiology and inflammation between these 16 and the other participants were. You may want to consider revising the manuscript to show only the exercise results for these 16 participants as the CD4/CD8 ratio was your primary outcome.
Our response: The assumptions and indications of expert 1 are completely correct. For methodological reasons, the analysis of the immunological phenotype was only carried out on a subgroup of n=16. We agree with the reviewer and must take this fact more into account. Both, the methodology and the results refer to the deviation in the number of participants. On the advice of the reviewer, we also evaluated the cyotkines again only for the subjects that we characterized with respect to the T cells. There were no differences compared to the case in which we included all subjects. Due to the already limited power of the statistics (due to the 3:1 randomization), as well as the reduced number of immune analyses, we decided to reconsider the focus of the main outcome measure. Due to the explorative character, we would like to waive main outcome measure in order to avoid over interpretation. Therefore, the results section was revised. Therefore, with regard to the other measures, we would like to keep the full sample size in order to obtain a better power at least for these. Nevertheless, immune analysis among a subgroup with small sample size can be seen as a further limitation as indicated in the discussion of the revised manuscript (lines 421-423).
- Please clarify how many of the total participants in each group had either a pathology or not. It looks like 30 people in the TG and 10 in the CG had a pathology, but it is unclear if this means everyone had a pathology or did some people have more than one pathology. In addition, the same is required for medications. Depending on how many participants in the whole study had a pathology you may need to describe differences between those with a pathology and those without at baseline.
Our response: We agree with the reviewer. The presentation in table1 could lead to confusion. We have revised it, added the number of healthy subjects, and added the number of subjects with one or more diseases/ taking medication to Table 1. Since most of the participants had a pathology, we cannot perform a robust analysis of the subgroups. The number of participants without pathologies or medication intake in the control group seems to be too small. The same applies to the subgroup of the immune analysis. To control for possible differences in health status between the two groups, baseline measurements were taken into account and, if necessary, an ANCOVA was performed as describe in line 234-238.
- How were the groups stratified for randomization and who performed the randomization?
Our response: All subjects who met the inclusion criteria were randomly assigned to the control or intervention group. We added this information in the methods part (lines 95-96).
- More details are required for the blood draws please.
- Were participants fasted before each blood draw, and were lifestyles such as diet and exercise controlled in the 24-hour period before blood draws?
Our response: Subjects were asked not to exercise for 24 hours prior to blood sampling, to eat as usual, and to arrive fasting in the morning for blood sampling. We added this information in the methods part (lines 175-176).
- It is unclear how the absolute T-cell counts were calculated, and to be clear was it the absolute counts or the percentages of CD4 and CD8 T-cells that were used to calculate the CD4/CD8 ratio?
Our response: We added this in Line 199 – 201 (The gating strategy used to identify T cell subsets was set as followed: at first, lymphocytes were gated on a forward scatter/side scatter (FSC/SSC) dot plot following gating on CD4 and CD8. In this gate, absolute count of CD4+ and CD8+ was performed to determine CD4+/CD8+ ratio.
- If the absolute counts were calculated as a percentage from a complete blood count this needs to be described. Also, if a CBC was completed it would be useful to include the counts of total WBCs, neutrophils, monocytes and lymphocytes.
Our response: Due to local conditions, we were unable to obtain a complete blood count of the subjects. The cells were counted only by means of a flow cytometer, and there we recorded only the corresponding subpopulations in terms of numbers. We have taken this into account as a limitation in the discussion (lines 421-423).
- At T2 there is a large mean increase, albeit non-significant (p=0.088), in absolute CD4+ T-cell counts and a small mean increase for absolute CD8+ T-cell counts for both the TG and CG. This is the likely cause of the change in the ratio. However, the changes in frequencies of T-cell subsets do not reflect these differences. Specifically, although the CD4/CD8 ratio increases there is no beneficial change in the CD4 or CD8 subset frequencies. Therefore, could the observed change in CD4/CD8 for the TG be because of absolute numerical changes in other cells that express CD4 (e.g. dendritic cells) and CD8 (e.g. NK-cells, MAITs, DCs) within the lymphocyte pool? Critically, because you have not used CD3 to identify T-cells, you may have inadvertently included other cells – this should be discussed.
Our response: Thank you very much for the analyses of the changes of CD4+/CD8+ ratio. We agree with this and discussed it in more detail in lines 326 - 331. “Thus, the changes in the T cell subpopulation do not offer an indication of the increased CD4+/CD8+ ratio. However, as we have a slight but not significant increase in absolute CD4+ and CD8+ cells, other reasons could be possible. Since we do not used CD3+ antibody to identify T cells, the change in CD4+/CD8+ ratio for the TG could because of absolute numerical changes in other cells that express CD4+ and CD8+ within the lymphocyte pool. This must be taken into account by interpretation of results.”
- For the cytokine analyses, please report the lower limit of detection (LLOD) and what percentage of each cytokine was within the LLOD
Our response: The assay ranges are as follows: CC-chemokine ligand (CCL) 2 (30.9 – 7500pg/ml), CCL13 (4.94 – 1200pg/ml), intercellular adhesion molecule 1 (ICAM-1) (7000 – 1700000pg/ml), IL (interleukin) -1ra (28.8 – 7000pg/ml), IL-2 (0.64 – 28.00pg/ml), IL-6 (0.854 – 35.00 pg/ml), IL-8 (1.6 - 11.70pg/ml), IL-10 (0.24 – 10.0 pg/ml), IL-18 (18.5 - 4500 pg/ml), leptin (494 - 120000 pg/ml), resistin (53.5 – 13000pg/ml), TNF-alpha (8.23 - 2000 pg/ml), and vascular endothelial growth factor (VEGF) (8.23 – 2000pg/ml). All measured values for the cytokines were within the assay range and the corresponding standard curve. We added this information in the methods part (lines 215-221).
- Given my above comments, the limitations and conclusion section may need revised.
Our response: We have included the relevant aspects into the limitations.
Grammar – there are a few errors throughout that I ‘m sure the journal will pick up, but to help I have highlighted a few.
Line 223. p=0,241 needs changed to p=0.241 - done
Line 319. State how many “a large proportion” is. We defined it in more detail.
Line 322. Change sports programmes to exercise programmes - done
Line 371. Change discuss to discussed - done
Table 2. Change mas to mass - done
Throughout, aging and ageing have both been used – please be consistent. - done
Reviewer 2 Report
There is an interesting study that one of its main novelties is focused on the characteristic of an exercise program. The authors introduce very well the study and their justification. However, the exercise program is not completely described. Please, described it in section method. In my opinion, the sample is not enough due to short number of participants in CG and either the heterogeneity of the two groups regarding with their pathologies and the drugs consumption.
As indicated by the authors, it is necessary to know the exercise activity in the control group in order to explain some results.
In regards with table 4 it could be interesting to analyse the differences of the differences. I mean, have the authors compared the increment or descend of selected cytokine plasma levels between CG and TG?
Also I miss the explanation of the results in IL-18 levels. Please, insert it in the Discussion section.
Author Response
Reviewer 2
- However, the exercise program is not completely described. Please, described it in section method.
Our response: We describe it in more detail in the methods part (lines 155 – 162).
- In my opinion, the sample is not enough due to short number of participants in CG and either the heterogeneity of the two groups regarding with their pathologies and the drugs consumption.
Our response: We cannot say with certainty that the group size of CG is large enough, as no sample size calculation was done. We have added this point in the limitations (line 422). Due to the explorative approach, we have also used a heterogeneous sample. We are also aware of limitations in this case, which we have explained in lines 423-424. However, we are convinced that the available data can provide an important indication for future studies. We decided on this design because of two reasons. The most important point is the following: The study was started after the first lockdown in Germany. Due to the requirements of the German government, contact persons of Covid-19 patients were quarantined for 14 days. Due to the 6-week intervention phase, a quarantine would have led directly to study exclusion. Since the risk was not to be underestimated from our point of view, we decided to increase the number of participants in case of drop-out due to quarantine. This is quite a common procedure when an increased drop-out to be expected. Secondly, we decided it to present a greater incentive for the subjects. The intention was to complete the study quickly before another lockdown could stop the study untimely. Subjects who are interested in this study type favor the intervention group accordingly. Experience has shown that the probability of being drawn into the intervention group increases the motivation to participate. We have listed this rationale in lines 97-102
- As indicated by the authors, it is necessary to know the exercise activity in the control group in order to explain some results.
Our response: This statement was speculative and cannot be verified. Therefore, we delete this statement. Due to ethical reasons, we did not restricted daily leisure-time physical activity. However, wearing activity tracker could be a possible reason to evaluate activity during study. We added this in our discussion in lines 405 – 407.
- In regards with table 4 it could be interesting to analyse the differences of the differences. I mean, have the authors compared the increment or descend of selected cytokine plasma levels between CG and TG?
Our response: Yes, we analyzed differences between TG and CG by interaction effect (Time* Group) of ANOVA or by, in case of non parametric testing, Mann-Whitney U-test, using the mean differences of T2 minus T1 values (described in lines 232-234). As shown, we detected no differences between the two groups.
- Also I miss the explanation of the results in IL-18 levels. Please, insert it in the Discussion section.
Our response: We analyzed IL-18 because of its associated to multiple components of the metabolic syndrome. It was also shown that it is affected by exercise. However, most studies which found a decrease of IL-18 after an exercise intervention used longer intervention times. We added this in the discussion (lines 394-397). (e.g. Trøseid M, Lappegård KT, Mollnes TE, Arnesen H, Seljeflot I. The effect of exercise on serum levels of interleukin-18 and components of the metabolic syndrome. Metab Syndr Relat Disord. 2009 Dec;7(6):579-84.).
Reviewer 3 Report
The manuscript titled “Effects of a 6-week …” reports findings of a study intended to reveal adjustments made by the immune system to a short period (5 weeks) of exercise training of moderate intensity, but recruiting all major muscle groups among aged individuals who may, or may not, have age-related morbidities (e.g. diabetes, obesity), and who may, or may not, show evidence of immunosenescence, or inflammaging. On the positive side, this is a meaningful study many in our aging population suffer from a blunting in the effectiveness of the immune system to fend off infections carried by pathogens. Another positive is the design of the training intervention which, reasonably, could be carried out by most of our aging population. The analyses feature tried and true procedures providing confidence in the results gathered. The overall message is that this sort of modest training regimen significantly improved the ratio of naïve (CD-4) cell sto memory (CD-8) cells suggesting greater immunological defensive capacity among and aged group of subjects. Moreover, the profile of circulating cytokines was also improved as a result of the training program, again suggesting enhanced defensive capacity against pathogen exposure. However, there are some issues that need to be adequately addressed by the authors.
General comments
1) Although it is stated on line 79 that there is precious little data regarding the effects of resistive strength training on the immune aging process, this shortcoming was not directly addressed here. Rather than combining strength and endurance training in the same regimen, those two modes of exercise should have been included separately to gain an increased knowledge base.
2) The authors (in line 82) speak as if it can be assumed that the subjects all possess an inflammatory disease (solely because of their age?). But this was not one of the inclusion criteria used for subject selection. Accordingly, such comments must be confined to the effects of aging and not inflammatory disease.
3) Sample size between groups was vastly different (30 vs. 10); please justify this and how this may have affected statistical analyses.
4) On lines 210-212, it is stated that due to the “explorative nature” of this study, no adjustments in alpha level were made when using multiple comparison testing. Why was a simple post-hoc measure, which would have held constant the alpha level throughout multiple comparisons, not used?
5) On lines 336-339, it is stated that it is possible that strength gains seen among the control group could be due to their increased physical activity during the 6 week intervention. If this is true, then all data from these control subjects must be discarded as they could not be truly considered controls.
6) Again, in lines 385-386, it is inferred that subjects studied here have pre-existing pathologies. Was this, or was this not, an inclusion criterion? If not, such statements as made here must be deleted.
Specific comments
Line
71 Strike “only”.
137 Is there a missing line below here, as it does not seem to be joined to line 138?
271 Chane to “Statistically significant findings …”
293 Delete “in” after “within”.
352 Delete “have”.
356 Consider replacing “evaluation” with “training”.
384 Replace “implicate” with “demonstrate”.
386 Delete “and” before ”individuals”.
Author Response
Reviewer 3
- Although it is stated on line 79 that there is precious little data regarding the effects of resistive strength training on the immune aging process, this shortcoming was not directly addressed here. Rather than combining strength and endurance training in the same regimen, those two modes of exercise should have been included separately to gain an increased knowledge base.
Our Response: We agree with this point. However, the intention was to evaluate the effects of a typical fitness workout, which is typically undertaken in a gym. We do not contribute to the mechanics of strength or endurance training. However, we are confident that the results of this study will have practical relevance for the peer group. We also hope that this is clear from the question and is made clear in the introduction.
- The authors (in line 82) speak as if it can be assumed that the subjects all possess an inflammatory disease (solely because of their age?). But this was not one of the inclusion criteria used for subject selection. Accordingly, such comments must be confined to the effects of aging and not inflammatory disease.
Our Response: We agree with reviewer 3. This sentence could lead to misunderstandings, why we have deleted it.
- Sample size between groups was vastly different (30 vs. 10); please justify this and how this may have affected statistical analyses.
Our Response: We decided on this design because of two reasons. The most important point is the following: The study was started after the first lockdown in Germany. Due to the requirements of the German government, contact persons of Covid-19 patients were quarantined for 14 days. Due to the 6-week intervention phase, a quarantine would have led directly to study exclusion. Since the risk was not to be underestimated from our point of view, we decided to increase the number of participants in case of drop-out due to quarantine. This is quite a common procedure when an increased drop-out to be expected. Secondly, we decided it to present a greater incentive for the subjects. The intention was to complete the study quickly before another lockdown could stop the study untimely. Subjects who are interested in this study type favor the intervention group accordingly. Experience has shown that the probability of being drawn into the intervention group increases the motivation to participate.
We have listed this rationale in lines 97-102.
This study design is possibly problematic with regard to reduced power. We have listed this in the limitation section – lines 426-428.
- On lines 210-212, it is stated that due to the “explorative nature” of this study, no adjustments in alpha level were made when using multiple comparison testing. Why was a simple post-hoc measure, which would have held constant the alpha level throughout multiple comparisons, not used?
Our response: This statement could have been misunderstood. We wanted to express that no further adjustment of the alpha level was made across all measured parameters. When measuring within a parameter, a post hoc analysis according to Bonfferoni was always applied. We will revise this statement.
- On lines 336-339, it is stated that it is possible that strength gains seen among the control group could be due to their increased physical activity during the 6 week intervention. If this is true, then all data from these control subjects must be discarded as they could not be truly considered controls.
Our response: This statement was speculative and cannot be verified. Therefore, we would delete this statement. However, wearing activity tracker could be a possible reason to evaluate activity during study. We added this in our discussion in lines 404 – 407 The leisure-time physical activity could also have an influence on the results of CG. Since we have not restricted them, at least it could be monitored in future studies.
- Again, in lines 385-386, it is inferred that subjects studied here have pre-existing pathologies. Was this, or was this not, an inclusion criterion? If not, such statements as made here must be deleted.
Our response: We agree with the reviewer and delete this statement. Existing pathologies was not an inclusion criterion. We did this in order to include subjects who, according to the average age, already had pre-existing conditions. The project was based on the idea to correspond as closely as possible to the character of health care research, which is as close as possible to the reality of people's lives.
Specific comments
Line
71 Strike “only”. - done
137 Is there a missing line below here, as it does not seem to be joined to line 138? - done
271 Chane to “Statistically significant findings …” - done
293 Delete “in” after “within”. - done
352 Delete “have”.- done
356 Consider replacing “evaluation” with “training”. -done
384 Replace “implicate” with “demonstrate”. - done
386 Delete “and” before ”individuals” - done
Round 2
Reviewer 1 Report
The authors have answered my questions and requests, the paper is much improved, and I look forward to seeing it published. Only a few additional revisions are required.
- Because of the change in primary outcome, which is acceptable, the abstract should be revised to reflect the differences in participant numbers assessed for immune cell measures.
- The absolute counts require a reference for the methodology with a brief description of whether counting beads were used or not. Please also state how many lymphocytes (e.g. 10,000, 20,000 etc) were acquired in what volume of buffer to determine the absolute counts. Table 3 absolute counts also need a unit of measurement (e.g. cells/uL).
- Please include at line 345 your comment to point 6 I raised – “All measured values for the cytokines were within the assay range and the corresponding standard curve.”
- It is unclear why you have added the #p-value of ANCOVA analyses in Table 3 for only CD4+ TEMRA and CD28-/CD57+ cells? The stats section suggested you completed a repeated measures ANOVA and not an ANCOVA on change scores. Having this for only two variables within the analyses is confusing. I would recommend keeping the analyses consistent throughout. Alternatively, you need to explain in the stats section why you have chosen to do these analyses on only these two cell subtypes.
Author Response
Reviewer 1
- Because of the change in primary outcome, which is acceptable, the abstract should be revised to reflect the differences in participant numbers assessed for immune cell measures.
Our response: We added this information in the abstract.
- The absolute counts require a reference for the methodology with a brief description of whether counting beads were used or not. Please also state how many lymphocytes (e.g. 10,000, 20,000 etc) were acquired in what volume of buffer to determine the absolute counts. Table 3 absolute counts also need a unit of measurement (e.g. cells/uL)
Our response: Our Cytoflex system is calibrated with calibrator beads of known concentration. The absolute number of cells was calculated by using the ratio of the number of cells observed in the respective gates. The buffer volume was 200ul and the absolute counts were about 10.000 cells. We added the unit information in table 3.
- Please include at line 345 your comment to point 6 I raised – “All measured values for the cytokines were within the assay range and the corresponding standard curve.”
Our response: We included the comment accordingly. Thank you.
- It is unclear why you have added the #p-value of ANCOVA analyses in Table 3 for only CD4+ TEMRA and CD28-/CD57+ cells? The stats section suggested you completed a repeated measures ANOVA and not an ANCOVA on change scores. Having this for only two variables within the analyses is confusing. I would recommend keeping the analyses consistent throughout. Alternatively, you need to explain in the stats section why you have chosen to do these analyses on only these two cell subtypes
Our response: Reviewer 1 is right. We deleted the ANCOVA analysis because there was no additional value of the calculations.